# FEATURE ACCENTUATION: REVEALING 'WHAT' FEATURES RESPOND TO IN NATURAL IMAGES

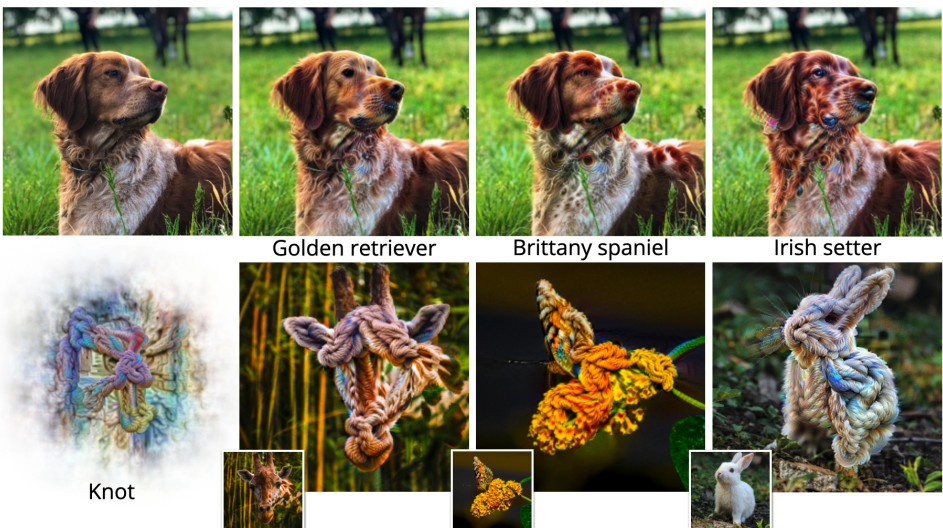

Figure 1: **Illustration of Feature Accentuation.** We introduce a novel method for *Feature Accentuation*, enhancing feature interpretability within images. Diverging from conventional feature visualization approaches, *Feature accentuation* operates with image-seeded inputs enabling local explanations. (**Top**) Our method can generate perturbations that accentuate a specific class or other logits. (**Bottom**) More generally, we propose to accentuate neurons or direction to understand model representations, (e.g. here, a knot neuron). Notably, this method operates *without any auxiliary generative models or reliance on pretrained robust models* and integrates with any pretrained classifier, offering a novel interpretability method.

## ABSTRACT

Efforts to decode neural network vision models necessitate a comprehensive grasp of both the spatial and semantic facets governing feature responses within images. Most research has primarily centered around attribution methods, which provide explanations in the form of heatmaps, showing *where* the model directs its attention for a given feature. However, grasping *where* alone falls short, as numerous studies have highlighted the limitations of those methods and the necessity to understand *what* the model has recognized at the focal point of its attention. In parallel, *Feature visualization* offers another avenue for interpreting neural network features. This approach synthesizes an optimal image through gradient ascent, providing clearer insights into *what* features respond to. However, feature visualizations only provide one global explanation per feature; they do not explain why features activate for particular images. In this work, we introduce a new method to the interpretability tool-kit, *feature accentuation*, which is capable of conveying both *where* and *what* in arbitrary input images induces a feature's response. At its core, feature accentuation is image-seeded (rather than noise-seeded) feature visualization. We find a particular combination of parameterization, augmentation, and regularization yields naturalistic visualizations that resemble the seed image and target feature simultaneously. Furthermore, we validate these accentuations are processed along a natural circuit by the model. We make our precise implementation of *Feature accentuation* available to the community as the *Faccent* library, an extension of *Lucent* (Kiat, 2020).

# 1 INTRODUCTION

Deciphering the decisions made by modern neural networks presents a significant ongoing challenge. As the realm of machine learning applications continues to expand, there is an increasing demand for robust and dependable methods to explain model decisions Doshi-Velez & Kim (2017); Jacovi et al. (2021). Recent European regulations, such as the General Data Protection Regulation (GDPR) Kaminski (2021) and the European AI Act Kop (2021), underscore the importance of assessing explainable decisions, especially those derived from algorithms.

In the domain of vision models, a variety of explainability methods have already been proposed in the literature (Simonyan et al., 2013; Selvaraju et al., 2017c; Smilkov et al., 2017; Sundararajan et al., 2017; Shrikumar et al., 2017; Zeiler & Fergus, 2014a;b; Springenberg et al., 2014a; Nguyen et al., 2015; Olah et al., 2017; Fel et al., 2021; Novello et al., 2022). One prominent category of these methods, known as *attribution*, generates heatmaps that highlight significant image regions in influencing a model's decision. The heatmap can be at the level of individual pixels based on their gradients (Simonyan et al., 2013; Bach et al., 2015; Baehrens et al., 2010; Smilkov et al., 2017; Sundararajan et al., 2017; Zeiler & Fergus, 2014b; Springenberg et al., 2014b; Srinivas & Fleuret, 2019; Yang et al., 2020; Montavon et al., 2017), or a coarse-grained map based on intermediate network activations/gradients (Zhou et al., 2016; Selvaraju et al., 2017b; Chattopadhay et al., 2018; Bae et al., 2020; Zhou et al., 2018; Wang et al., 2020; desai & Ramaswamy, 2020; Fu et al.; Kim et al., 2021) or input perturbations (Fel et al., 2021; Novello et al., 2022; Fong & Vedaldi, 2017; Zintgraf et al., 2017; Petsiuk et al., 2018; Fel et al., 2023b; Ribeiro et al., 2016; Lundberg & Lee, 2017), or even a combination of fine-grained and coarse-grained attribution (Selvaraju et al., 2017b; Rebuffi et al., 2020).

These methods have a wide range of applications, including improving decision-making, debugging, and instilling confidence in model outputs. However, they have notable limitations and drawbacks, as highlighted in several articles (Kindermans et al., 2019; Adebayo et al., 2018a; Slack et al., 2020; Ghalebikesabi et al., 2021; Fel et al., 2023c; Kim et al., 2022; Hase & Bansal, 2020; Nguyen et al., 2021; Shen & Huang, 2020; Sixt et al., 2022). These issues range from the problem of confirmation bias – just because an explanation makes sense to a human doesn't necessarily mean it reflects the inner workings of a model – to the more significant limitation of these methods – they reveal *where* the model is looking but not *what* it has observed. As a result, the research community is actively exploring new techniques for conveying the *what* aspect of model behavior.

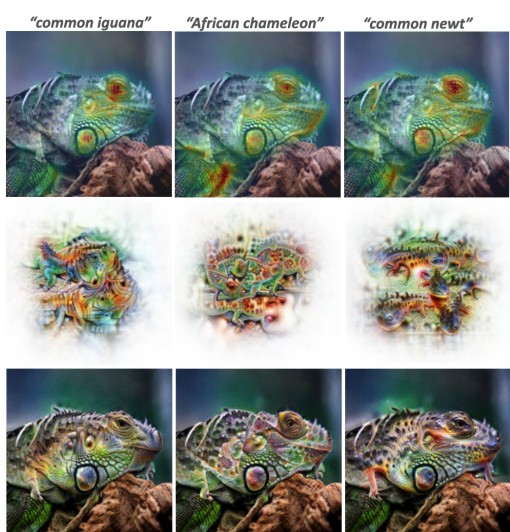

Figure 2: An iguana excites several class logits in InceptionV1, but what about the image excites each logit? Attribution maps highlight important regions of the image, but not *what* each logit sees in the region. Feature visualizations yield an exemplar for each logit, but these are hard to relate to the iguana image. Feature accentuation (ours) constitutes a powerful intermediary, transforming the iguana into a local exemplar for each class.

In response to these challenges, feature visualizations emerge as a compelling solution. They create images that strongly activate specific neurons or neuron groups (Szegedy et al., 2013; Nguyen et al., 2015; Olah et al., 2017; Ghiasi et al., 2021; 2022). The simplest feature visualization method involves a gradient ascent process to find an image that maximizes neuron activation. However, this process can produce noisy images, often considered adversarial, without some form of control. To address this issue, various regularization techniques (Olah et al., 2017; Mahendran & Vedaldi, 2015; Nguyen et al., 2015; Tyka; Audun) and data augmentations (Olah et al., 2017; Tsipras et al., 2018; Santurkar et al., 2019; Engstrom et al., 2019; Nguyen et al., 2016b; Mordvintsev et al., 2015) have been proposed, along with the use of generative models (Wei et al., 2015; Nguyen et al., 2016a; 2017).

Despite the promise of these methods, recent research has pointed out their limitations and potential pitfalls (Zimmermann et al., 2021; Geirhos et al., 2023). It is argued feature visualizations may be

misleading, as they can trigger intermediate network features that differ significantly from natural images; i.e. they produce neuron activation by way of a different circuit. Additionally, a given neuron may return high activation for a broad range of images, many of which cannot be easily related to the exemplars generated with feature visualization. In many practical settings, it is necessary to explain why a neuron responds to some *particular natural image*, such as an image that is misclassified. Thus in practice attribution methods see far more use than feature visualizations, given they provide image conditional explanations.

A final family of techniques that could serve to explain these difficult edge cases are visual counterfactual explanations (VCEs) (Goyal et al., 2019; Poyiadzi et al., 2020; Verma et al., 2020). VCEs attempt to specify an image $x'$ close to $x$ that explains '*how would $x$ need to change for the model to consider it an instance of class $y$?*' Contemporary VCE methods utilize variations of image-seeded (rather than noise-seeded) feature visualization, thus could potentially be used to explain *what* excites a feature in a particular natural image. However, visually compelling VCEs have only been synthesized in this way using either adversarially trained models (Boreiko et al., 2022; Gaziv et al., 2023; Madry et al., 2018) or by optimizing under the guidance of an auxiliary generative model (Augustin et al., 2022). Thus far in the literature, image-seeded feature visualizations of non-robust models under self-guidance yield one of two results, neither of which constitute a useful explanation. 1) When optimization is unconstrained, unnatural *hallucinations* appear across the entire image, similar to those popularized by *deepdream* (Mordvintsev et al., 2015). 2) When the optimization is constrained to remain close to the seed image, the synthesized image does not change in a perceptually salient way, as with classic adversarial attacks (Szegedy et al., 2013; Goodfellow et al., 2014; Moosavi-Dezfooli et al., 2016; Akhtar & Mian, 2018).

In the present work we introduce *Feature Accentuation*, a novel implementation of image-seeded feature visualization that does not require robust/auxiliary models. Feature accentuation can provide diverse explanations across both neurons and the images that activate them. Specifically, in this article:

- We formally introduce Feature Accentuation, a new post-hoc explainability method that reveals both *where* and *what* in natural images induces neuron activation.

- We show for the first time that high quality VCEs can be generated for non-robust models with image parameterization, augmentation, and regularization alone, without the use of an auxilary generative model.

- We show that feature accentuations drive neurons through the same circuits as natural images, which is not true of conventional feature visualizations. In fact, feature accentuations follow a class logit's prototypical circuit *more closely* than natural images do!

- We demonstrate the diverse range of applications feature accentuation facilitates.

- We release the open-source *Faccent* Library, an extension of the *Lucent* (Kiat, 2020) feature visualization library for generating naturalistic accentuations.

## 2 METHODS

**Notation.** Throughout, we consider a general supervised learning setting, with an input space $\mathcal{X} \subseteq \mathbb{R}^{h \times w}$, an output space $\mathcal{Y} \subseteq \mathbb{R}^c$ and a classifier $f : \mathcal{X} \to \mathcal{Y}$ that maps inputs $x \in \mathcal{X}$ to a prediction $y \in \mathcal{Y}$. Without loss of generality, we assume that $f$ admits a series of $L$ intermediate spaces $\mathcal{A}_\ell \subseteq \mathbb{R}^{p_\ell}, 1 < \ell < L$. In this setup, $f_\ell : \mathcal{X} \to \mathcal{A}_\ell$ maps an input to an intermediate activation $a = (a_1, ..., a_{p_\ell})^\top \in \mathcal{A}_\ell$ of $f$. We respectively denote $\mathcal{F}$ and $\mathcal{F}^{-1}$ the 2-D Discrete Fourier Transform (DFT) on $\mathcal{X}$ and its inverse. In this article, a feature denotes a vector in a feature space $v \in \mathcal{A}_\ell$ and its associated feature detector function is denoted as $f_v(x) = f_\ell(x) \cdot v$. Finaly, $|| \cdot ||$ denotes the $\ell_2$ norm over the input and activation space.

**Feature Visualization.** Activation maximization attempts to identify some prototypîcal input $x^* = \arg\max_x f_v(x)$ for any arbitrary feature detector function $f_v$ of interest (e.g. a channel or a logit of a convolutional neural network). To generate meaningful images, as opposed to noisy ones, a variety of techniques are typically employed, including some that we will also adopt in our approach. First and foremost, the choice of image parametrization plays a critical role, which is typically not pixel space but the frequency domain $\mathcal{Z} \subseteq \mathbb{C}^{h \times w}$, which give us $x = \mathcal{F}^{-1}(z)$. This parameterization affords precise control over the image's frequency components, facilitating the removal of adversarial high-frequencies. Additionally, this re-parametrization has the effect of

modifying the basins of attraction during optimization, yielding images that exhibit a more natural appearance. A second commonly employed strategy is the application of transformations at each gradient step to enhance image robustness. Noise injection or localized image optimization through cropping are frequently employed in this context. Formally, we denote $\mathcal{T}$ as the set of possible transformations, and $\boldsymbol{\tau} \sim \mathcal{T}$ represents a transformation applied to $\boldsymbol{x}$ such that $\boldsymbol{\tau}(\boldsymbol{x}) \in \mathcal{X}$. Putting everything together, for a feature detector $\boldsymbol{f_v}$, we associate the optimal feature visualization $\boldsymbol{z}^*$:

$$\boldsymbol{z}^* = \operatorname*{arg\,max}_{\boldsymbol{z} \in \mathbb{C}^{h \times w}} \mathcal{L}(\boldsymbol{z}) \;\; with \;\; \mathcal{L}(\boldsymbol{z}) \triangleq (\boldsymbol{f_v} \circ \boldsymbol{\tau} \circ \mathcal{F}^{-1})(\boldsymbol{z})$$

However, this feature visualization method presents several challenges: it lacks image specificity and therefore fails to provide local explanations (for a given point). Most of the time, it converges to a perceptually similar optima, providing only one visual explanation of the feature detector. Finally, as we will explore in the Section 3, Geirhos et al. (2023) observe that the generated images follow circuits which are very different from the circuits followed by natural images, thus raising questions about the validity of the explanations. We address these issues by introducing *Feature accentuation*.

## 2.1 FEATURE ACCENTUATION

We begin by adapting feature visualization through seeding, i.e., initializing this process at $\boldsymbol{z}_0 = \mathcal{F}^{-1}(\boldsymbol{x}_0)$, where $\boldsymbol{x}_0$ represents the natural image under investigation. In addition to introducing diversity into the explanations – each image will have a really different accentuation, even for similar neuron maximization – this approach allows us to commence from a non-OOD (Out-Of-Distribution) starting point. However, there is no guarantee that during the optimization process the image will remain natural or be perceived as such by the model. Additionally, if one desires a local explanation of $\boldsymbol{f_v}(\boldsymbol{x}_0)$, it is possible the synthesized image is processed differently by the model than the seed image. To address these concerns, we will augment the loss function with a regularizer that encourages intermediate activation to be closer to the original one. Formally, we introduce *Feature accentuation* loss:

**Definition 2.1 (*Feature accentuation*)** *The feature accentuation results from optimizing the original image $\boldsymbol{x}_0$ such that:*

$$\boldsymbol{z}^* = \operatorname*{arg\,max}_{\boldsymbol{z} \in \mathbb{C}^{h \times w}} \underbrace{(\boldsymbol{f_v} \circ \boldsymbol{\tau} \circ \mathcal{F}^{-1})(\boldsymbol{z})}_{maximize\ feature\ activation} - \overbrace{\lambda || (\boldsymbol{f_\ell} \circ \boldsymbol{\tau} \circ \mathcal{F}^{-1})(\boldsymbol{z}) - (\boldsymbol{f_\ell} \circ \boldsymbol{\tau})(\boldsymbol{x}_0) ||}^{maintain\ proximity\ to\ the\ \textbf{transformed}\ original\ image}$$

In various image manipulation applications, the regularization term is often defined in terms of pixel space distance (Santurkar et al., 2019; Augustin et al., 2020; Boreiko et al., 2022). However, for our specific application, our concern is not the pixel space distance per se, but rather ensuring that the synthesized image is processed in a manner similar to the seed image by the model. Consequently, we explore the implications of regularizing based on distances measured in the latent layers of the model. Here, we draw inspiration from previous work on "feature inversion" (Mahendran & Vedaldi, 2015; Olah et al., 2017), in which an image is synthesized to possess the same latent vector as a target image. The choice of the layer for feature inversion significantly impacts the perceptual similarity between the synthesized image and the target. Earlier layers tend to produce synthesized images that are more perceptually similar, as these layers share more mutual information with the pixel input. Conversely, using later layers introduces differences in the synthesized image to which the higher-order representations are invariant. We note that a distinctive feature of our approach compared to previous works is that we apply the same augmentation $\boldsymbol{\tau}$ to both the target $\boldsymbol{x}_0$ and the accentuated image $\mathcal{F}^{-1}(\boldsymbol{z})$ at each iteration, ensuring a robust alignment with the original image. Figure 3 illustrates the impact of $\lambda$ on the feature accentuation process. A smaller value of $\lambda$ permits substantial alterations across the entirety of the image. In contrast, a larger $\lambda$ leads to accentuation that closely resembles the seed image. The choice of the layer for measuring distance also exerts an influence on the outcome. Traditional pixel space regularization tends to produce subtle, low-amplitude patterns distributed across the entire image. Conversely, regularization in the latent space of the model both constrains and enhances the accentuation to critical regions.

We find that applying regularization in the early layers of the model yields comparable and desirable outcomes, whereas regularization in later layers leads to distortions resembling those observed in prior work on feature inversion.

Now that we have established the fundamentals of our approach, the subsequent sections will explore two additional ingredients that make *Feature accentuation* shine; the image parametrization and augmentations.

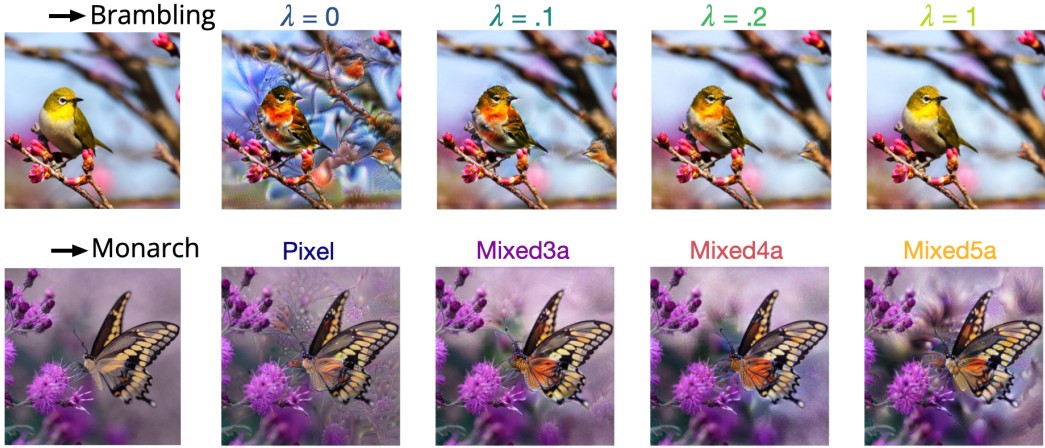

Figure 3: **(Top)** Influence of Regularization Strength ($\lambda$). With no regularization accentuations significantly deviate from the original image, while excessive regularization prevents meaningful alterations to the image. **(Bottom)** Influence of Regularization Layer ($\boldsymbol{f}_\ell$) – from pixel space ($\mathcal{X}$) to a deep layer of InceptionV1 (mixed5). Regularization in pixel space does not enable meaningful image modifications, whereas regularization in excessively deep layers produces hallucinations. Intermediate regularization accentuates existing image details that drive the logit without injecting new features across the entire image.

## 2.2 The Proper Parameterization

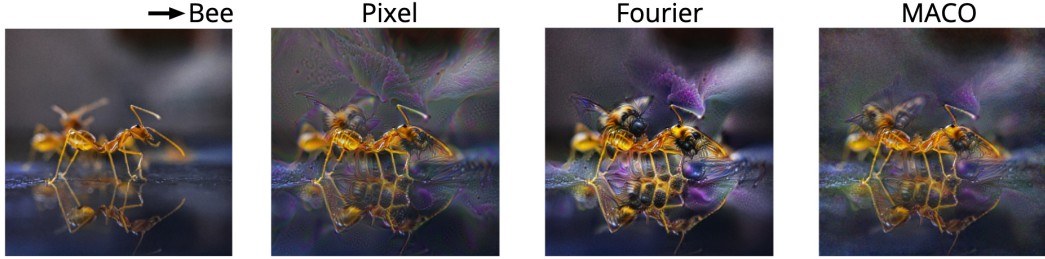

Figure 4: **Effect of image parameterization.** Modern parameterizations such as Fourier and MACO method enable a more effective management of high frequencies, thereby generating meaningful perturbations. For the remainder of this article, we choose the Fourier parameterization for the InceptionV1 model, although MACO can also be employed, albeit with greater constraints and suitability for deeper models.

We proceed with an examination of the quality of the images generated by *Feature accentuation* employing distinct image parameterizations. Our experimentation centers on the InceptionV1 network, which is well-established in the feature visualization literature. In this evaluation, we contrast three parameterizations of increasing complexity: *pixel*, *Fourier* (Olah et al., 2017; Mordvintsev et al., 2018), and *MACO*, introduced in Fel et al. (2023a).

The *Fourier* parameterization (Mordvintsev et al., 2018; Olah et al., 2017) is a well-recognized selection for feature visualization. It employs weight factors $\boldsymbol{w}$ to emphasize low frequencies and mitigate the optimization of high-frequency (adversarial) noise. This parameterization can be expressed as $\boldsymbol{x} = \mathcal{F}^{-1}(\boldsymbol{z} \odot \boldsymbol{w})$, where $\odot$ denotes the Hadamard product. The *MACO* parameterization, akin to *Fourier*, operates by representing the image in frequency domain, but in polar form $\boldsymbol{z} = \boldsymbol{r}e^{i\boldsymbol{\varphi}}$, thus allowing optimizing only the phase, $\boldsymbol{\varphi}$, while preserving the fixed magnitude spectrum to avoid introducing frequencies absent in natural images. This strict constraint on high frequencies contributes to the generation of more natural images, particularly in deep models. In the original work, $\boldsymbol{r}$ was set to match the average magnitude spectrum of the ImageNet training set (Deng et al., 2009). However, when applying *MACO* to feature accentuation, we initialize optimization with a target image $\boldsymbol{x}_0$, thereby constraining $\boldsymbol{r}$ to the magnitude spectrum of $\boldsymbol{x}_0$ and varying it for each image. Figure 4 presents an example of feature accentuation using the three different parameterizations. Notably, we

observe that the *pixel* parameterization tends to optimize for adversarial noise, as previously noted in the literature. In contrast, the alternative *Fourier* and *MACO* parameterizations introduce perceptible and semantically meaningful changes to the image. However, we find that the *MACO* parameterization produces slightly noisier and lower contrast visualizations; we believe in this context *MACO* is over-constrained. We will therefore use the Fourier parameterization for the rest of the article (see Appendix for more results).

## 2.3 THE APPROPRIATE AUGMENTATION

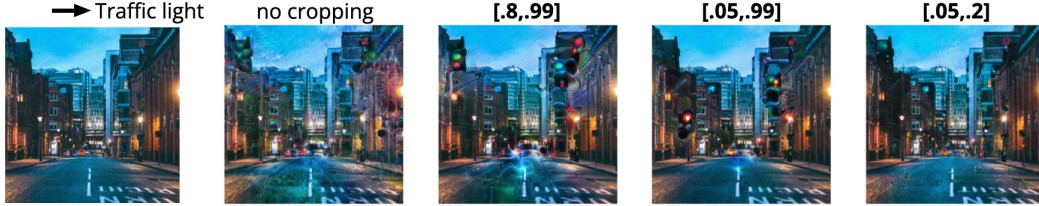

Figure 5: The effect of random **crop augmentations**. Square brackets indicate the `[minimum, maximum]` permissible dimensions (in %) for the bounding box crop. Smaller crops add definition to the image, but when only small crops are applied tiny hallucinations appear. We find applying the full range of crops each batch to produce balanced results.

Fel et al. (2023a) show that random cropping with gaussian and uniform noise are a simple yet effective set of augmentations for feature visualizations. Where the *MACO* crop augmentations were sampled with a center bias in the original work, we sample uniformly for feature accentuations. We apply 16 random transformations every iteration, averaging $\mathcal{L}$ over this batch. We find that incorporating smaller box crops into this augmentation scheme improves the definition of feature accentuations, as can be seen in Figure 5.

## 2.4 ADDING SPATIAL ATTRIBUTION

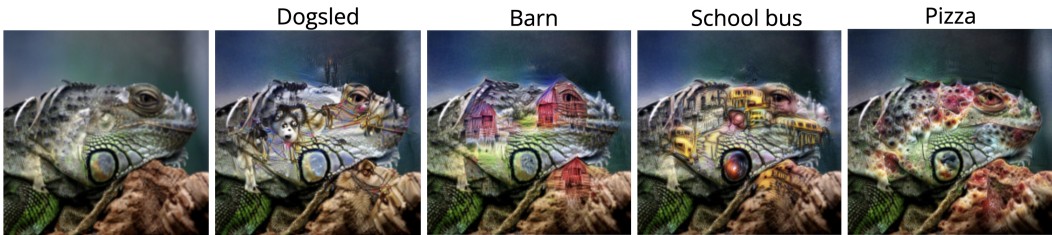

Figure 6: **Accentuating unrelated features in images can lead to significant changes.** Accentuations of the most inhibited logits for the iguana image. While the model doesn't associate any 'schoolbus-like' features with the iguana, an observer might mistakenly think otherwise due to the result of feature accentuation. To address this, we suggest incorporating spatial attribution in Section 2.4 as the final ingredient for *Feature accentuation*.

We have demonstrated that a combination of regularization, parameterization, and augmentation techniques can produce naturalistic enhancements of image features, which exhibit similarities to both the original seed image and the target feature. However, let us revisit the primary objective of our methodology: feature accentuations should elucidate *what* a feature responds to "within the seed image." What conclusions might one draw from accentuations in this regard? Intuitively, *Feature accentuation* maximizes activation by *exaggerating* important areas of the seed image. Thus, one might conclude the feature is present in the seed image in precisely those areas that undergo change.

Afterall, we have seen that with the correct implementation presumably unimportant regions of the image, such as the background, experience minimal perceptual changes when accentuated. However, it is crucial to observe that an image already exemplifying the target feature will also undergo no changes when accentuated, since it already represents a local optimum. Conversely, images unrelated to the target feature exhibit substantial changes upon accentuation. For instance, Figure 6

illustrates accentuations of the logits *most inhibited* by the iguana image. Notably, the model does not associate any "schoolbus-like" attributes with the iguana, but an observer might erroneously interpret otherwise based on the corresponding feature accentuation. In light of this consideration, we propose that the final component of *Feature accentuation* should encompass spatial attribution, thereby reintroducing *where* into the *what*.

As discussed in Section 1, numerous attribution methods have been introduced in the literature. In this article, we maintain a general perspective, not favoring any specific algorithm. However, for an attribution map—computed by any means—to align effectively with the objectives of feature accentuation, we propose a straightforward modification.

In the typical scenario, an attribution map is represented as a function $\varphi(\boldsymbol{f_v}, \boldsymbol{x}) \subseteq \mathbb{R}^{h \times w}$, where the values in the attribution map $\varphi(\boldsymbol{f_v}, \boldsymbol{x})$ delineate significant spatial regions in $\boldsymbol{x}$ that influence the activation produced by the feature detector. Visualizing an attribution as a heatmap necessitates normalization. Typically, normalization occurs with respect to the values in a single attribution map, rendering the intensity entirely contingent on the variance of $\varphi(\boldsymbol{f_v}, \boldsymbol{x})$. As a result, an image containing prominent features, such as bananas distributed extensively, might undergo normalization that diminishes the impact of these features, leading to an explanation that does not effectively capture their significance. Conversely, an image devoid of such features may experience a normalization process that artificially amplifies any minor attributes present. Consequently, both images can end up with similar explanations, despite substantial differences in their content and the features of interest (as shown in Figure 7).

To mitigate this issue and improve the fidelity of feature representations, we propose a "collective" normalization approach, which leverages the attribution values derived from a comprehensive set of natural images. This approach enables a more accurate assessment of feature relevance across diverse contexts, allowing for a finer-grained distinction between images with disparate feature compositions. Subsequently, these resulting maps can serve as opacity masks for the feature accentuation applied to the image. As illustrated in Figure 7, this normalization approach enables the possibility that an image may activate a feature throughout its entirety or not at all.

Figure 7: We introduce a normalization *across* images to provide a sensible mask for *Feature accentuation*.

## 3 EXPERIMENTS

**Circuit Coherence Assessment** *Feature accentuation* produces perceptually meaningful transformations, but if they are to constitute good explanations of a model's responses, it's important we validate they are processed by the model in a natural way. To assess this, we adopt an approach introduced in Geirhos et al. (2023), which measures the degree to which the paths (intermediate activations) taken by the images align with natural images of the studied class. Formally, to quantify this concept of "circuit similarity" between two images, $\boldsymbol{x}_i$ and $\boldsymbol{x}_j$, we measure the pearson correlation of their hidden vectors, $\rho(\boldsymbol{a}_{\ell,i}, \boldsymbol{a}_{\ell,j})$, and average across pairs of images, across all layers $\ell$ of the network. As depicted in Figure 8 (A), conventional feature visualization methods perform inadequately in this regard. Specifically, it has been demonstrated that for natural images of a class and his associated feature visualizations, there exists a low correlation across most layers of the network when compared to the correlation observed among natural images themselves. In contrast, *Feature accentuation* capitalizes on its image-seeded approach and regularization techniques to maintain a notably high level of naturalness in its internal pathways. It even achieves a super-natural score, signifying a superior inter-correlation with natural images than the correlation observed among natural images themselves.

**Effect of** $\lambda$**.** Based on these findings, it is pertinent to investigate the behavior of the pathway in function of the introduced regularization parameter, $\lambda$. Figure 8 (B) illustrates the effect of varying $\lambda$ on the naturalness of the pathway. Naturally, we observe that strong regularization compels the image path to remain close to the natural images, resulting in excellent scores for early layers but deteriorating scores in later layers. Conversely, moderate regularization allows us to maintain significantly higher scores compared to unregularized optimization, suggesting the existence of an

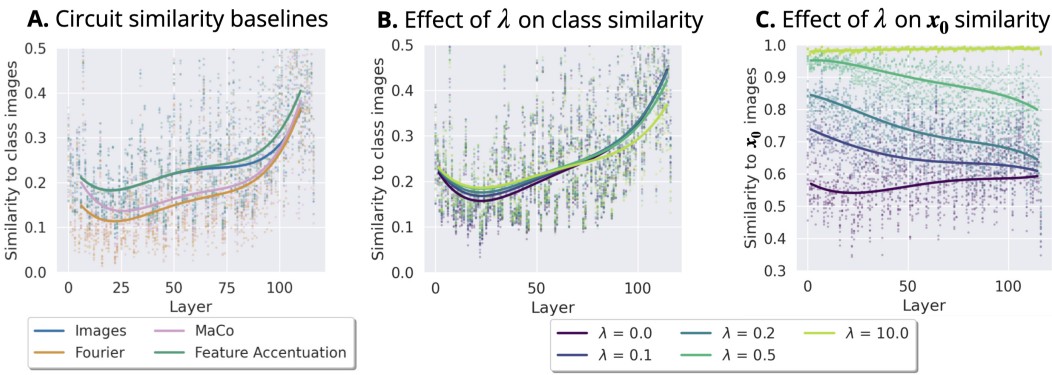

Figure 8: **Path similarity of Feature Accentuations. A.** We measure path similarity (correlation) of accentuations and natural images, as compared to the similarity of other synthesized/natural images. Unlike traditional feature visualization, *Feature accentuation* for a class follow an internal path that is similar to natural images of that class, and even closer on average to natural images than the natural images themselves, hence the term "supernatural images" **B.** We observe the impact of $\lambda$ on this measure, and **C.** how the path similarity to the seed image is influenced by $\lambda$. A high $\lambda$ value brings *Feature accentuation* closer to the original image, while a $\lambda$ value of zero results in deviation from naturalistic images towards something resembling feature visualization. This reveals a "sweet spot" for $\lambda$ in achieving supernatural images

optimal $\lambda$ value. Furthermore, in Figure 8 (C), we examine the correlation between the optimized image and, instead of the distribution of natural images, the seed image. As anticipated, a higher $\lambda$ corresponds to a higher correlation. It is noteworthy that, to achieve a strong score in class similarity, *Feature accentuation* does not necessarily adhere faithfully to the input image. Instead, it employs pathways that exist in other images.

**Misclassifications** Understanding why a model misclassifies inputs is one of the primary aims of explainability research. Figure 9 demonstrates how exaggerating the predicted logit with *Feature accentuation*can assist in diagnosing misclassifications, helping the user see the same hallucination as the model.

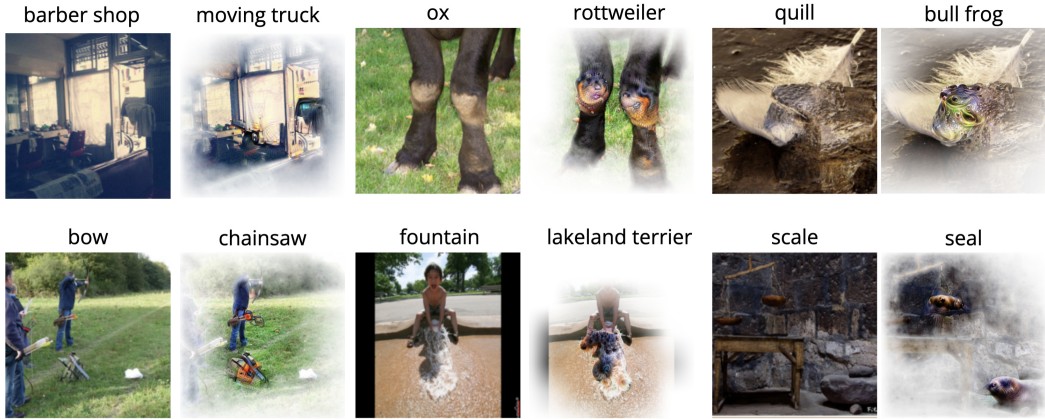

Figure 9: *Feature accentuation* **enables understanding of failure cases** by emphasizing incorrectly predicted classes, allowing the human user to share in the model's hallucination. For instance, the top left example shows a barber shop misclassified as a truck due to the shop's wrinkled curtain being mistaken for a truck's side, while the first example in the bottom row displays an archery image classified as a chainsaw because the orange-tipped arrows resemble a the handle, and the gray shafts resemble a saw blade.

**Explaining Latent Features** A comprehensive understanding of a neural network model must include its latent features in addition to its logits. However, explaining activations within the latent

space poses a greater challenge than explaining activations in the logit space due to the absence of direct correspondence between latent features and predefined semantic labels. Typically, latent features are probed using feature visualizations or dataset examples that elicit the highest activations. Feature accentuations present a novel approach for unveiling the multifaceted manifestations of a specific latent feature within natural images. Similar to conventional feature visualizations, this process can be executed with respect to individual neurons, channels, or even arbitrary feature directions, such as those associated with conceptual dimensions as in (Fel et al., 2022).

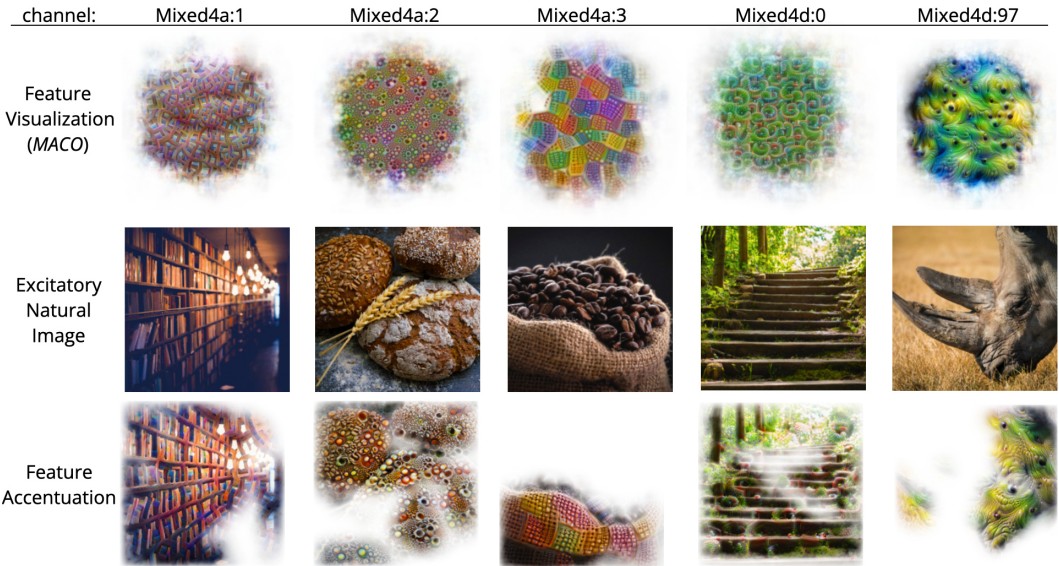

Figure 10: **Feature accentuation applied to latent channels** can reveal the multitude of ways features manifest within natural images. In contrast to conventional feature visualization, feature accentuation allows us to amplify the presence of targeted features in real-world images, granting insights into how they *behave in the wild*.

## 4 LIMITATIONS

We have demonstrated the generation of realistic feature accentuations for neural networks using an improved feature visualization technique. However, it is essential to recognize that creating realistic images does not automatically guarantee effective explanations of neural networks. Furthermore, to gain informative insights into the model, especially in identifying spurious features, feature visualizations might need to produce images that deviate from the original input, potentially conflicting with the proposed regularization. Previous research has also shed light on the limitations of feature visualizations, as observed in well-regarded studies Borowski et al. (2020); Geirhos et al. (2023); Zimmermann et al. (2021). A prominent critique concerns their limited interpretability for humans. Research has shown that dataset examples are more effective than feature visualizations for comprehending CNNs. This limitation could arises from the lack of realism in feature visualizations and their isolated application. We strongly advocate for the use of supplementary tools such as attribution maps and concept-based explanations alongside Feature Accentuation, to build a comprehensive understanding of neural networks.

## 5 CONCLUSION

Understanding why a model detects a particular feature in an image is a pivotal challenge for XAI. In this article, we introduce *Feature Accentuation*, a tool designed to exaggerate features in an image to aid our understanding. This results in meaningful and natural morphisms without resorting to GANs or robust models, thereby ensuring that the explanations provided by *Feature accentuation* are exclusively attributable to the scrutinized model. Our findings demonstrate that the generated images (i) undergo processing akin to natural images, (ii) can help comprehension of model failure cases and (iii) internal features. We anticipate that the utilization of *Feature accentuation* will pave the way for enhanced XAI methodologies and better understanding of the internal representation of discriminative vision models.

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

# APPENDIX FOR FEATURE ACCENTUATION

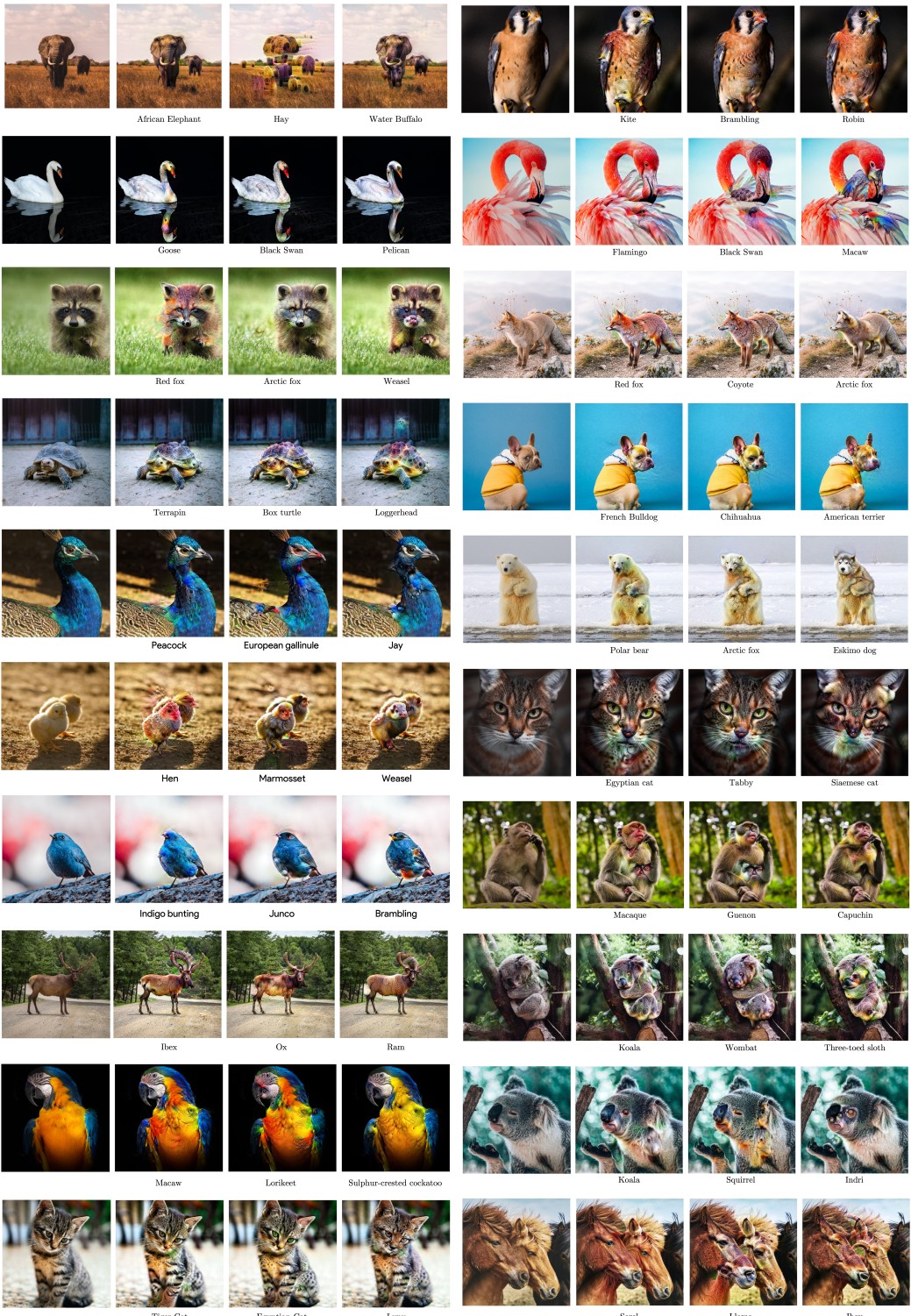

Figure 11: Additional Class-wise accentuations

## A  PATH EXPERIMENT DETAILS

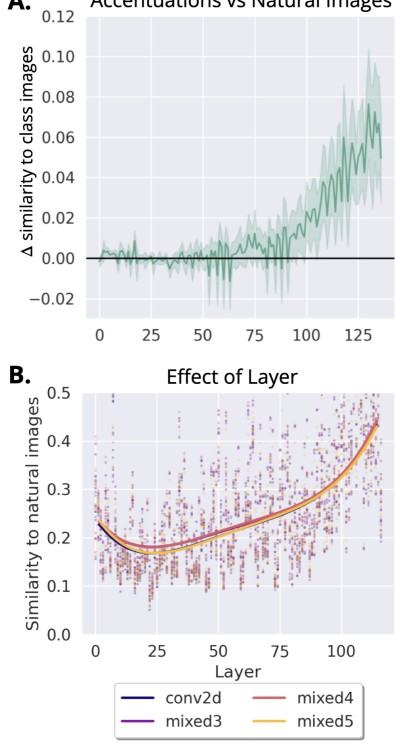

Figure 12: **A.** The path similarity of feature accentuations to natural images, normalized such that the zero-line corresponds to the path similarity of natural images to themselves. Banded region corresponds to the +/-1 standard deviation across 50 classes tested. **B.** Small effect of regularization layer on path coherence.)

For each of the 5 $\lambda$s tested (0,.1,.2,.5,10) we generated 1751 accentuations, corresponding to all the correctly predicted images across 50 random classes in the Imagenet (Deng et al., 2009) validation set. We accentuated each image toward its class label for 100 optimization steps using the Adam optimizer, with a .05 learning rate. We used 16 augmentations each optimizition step, cropping with a `maximum` box size of .99, and a `minimum` box size of .05. As an additional augmentation we add gaussian and uniform noise with $\sigma = .02$ (given initial images normalized to the range [0,1]) each iteration. We regularized through layer *mixed3a*, but note that the optimal regularization layer seems to interact with the layer of $f_v$. Besides this and $\lambda$, we observe the above hyperparameters produce quality accentuations in the general case (see section G).

Within a given class, we get the average pair-wise correlation for each natural image, and similarly the average correlation of each accentuation to the natural images (excluding the correlations between accentuations and their natural image pair, which would be trivially high). The curves depicted in Figure 8 are a spline interpolation (degree 2), across the underlying data-points (class-wise average correlations) averaged into 10 bins. We took this approach so as to convey the raw data and the general trend simultaneously. A difficulty with plotting this data stems from the fact that there is a large amount of variance in the correlation measure across nearby layers of different architectural type. We can filter out this variance in Figure 8.a and isolate the effect we are interested in by plotting the difference between the correlation measure for accentuations and natural images in each layer. This version of the plot can be seen in Figure 12.a. It requires no smoothing, but does not convey the absolute magnitude of the correlation across layers.

In addition to our experiments with $\lambda$, we tested the effect of regularization layer on the class-wise path similarity metric. We found only a small effect, with the earliest and latest layers tested (*conv2d0* and *mixed5a*) performing slightly worse, in agreement with our qualitative evaluation of the corresponding accentuations.

Figure 13 shows a random sample of our *super-natural* accentuations. These images are processed by way of hidden vectors that better correlate with those for natural class images than the natural images correlate with themselves.

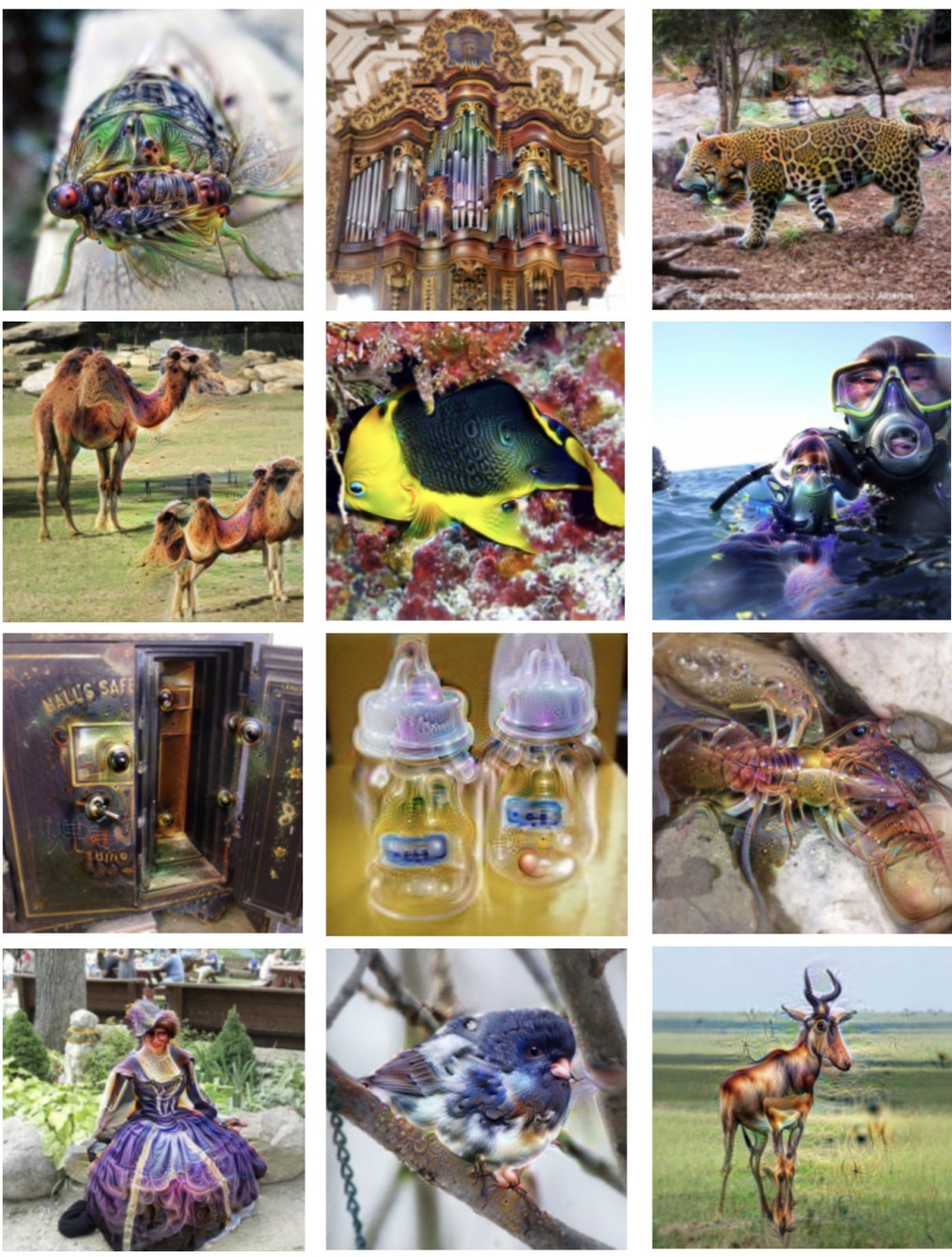

Figure 13: A sample of *'super-natural'* class images, which follow a prototypical path through the model (see section 3).

## B MASKING

Here we will briefly expound upon our normalized attribution method, put forward in section 2.4. For a data sample $X$ we consider a set of attribution maps $\{\varphi(f_v, x)\} \mid x \in X\}$. Let $v$ represent the flattened, ordered vector of every scalar attribution in this set. We can then choose a percentile range $(p_1, p_2)$, such that values outside the range are fully masked/fully visible in the upsampled mask, and values in between are partially masked. We can then normalize every element, $u$, of a particular attribution map, $\varphi(f_v, x)$, by;

$$\mathcal{N}(u; v, p_1, p_2) = \begin{cases} 0 & \text{if } u \leq P(v, p_1) \\ \frac{u - P(v, p_1)}{P(v, p_2) - P(v, p_1)} & \text{if } P(v, p_1) < u < P(v, p_2) \\ 1 & \text{if } u \geq P(v, p_2) \end{cases} \quad (1)$$

By setting $p_1$, the user can control how globally salient a region must be before it constitutes a partial expression of the feature and can be unmasked. Conversely, by setting $p_2$, the user determines the percentile past which a feature is 'fully expressed', and fully unmasked. As previously stated, this approach is agnostic to the attribution method used; for example we applied this normalization approach to *gradCAM* (Selvaraju et al., 2017a) attribution maps to generate the masks for Figure 9, but simply used the features activation map itself as our attribution for latent accents in Figure 10, as convolutional layers are already spatialized.

## C A NOTE ON LEARNING RATE

We observe learning rate that works for noise-seeded feature visualizations may be too large for feature accentuation, causing the visualization to deviate drastically from the target image in the the initial steps and never make it back. A smaller learning rate keeps the visualization perceptually similar to the seed image.

## D ADDITIONAL EXAMPLES FOR HYPERPARAMETERS

In the interest of conserving space, we initially utilized a single example image to illustrate the impact of manipulating hyperparameters that we consider pivotal for *Feature accentuation*. However, to bolster our demonstration, we present here a series of supplementary examples drawn from our comprehensive testing dataset, aimed at showcasing the consistent and robust nature of the described effects across various instances. We deliberately choose to show some examples multiples times, so the reader can get a sense for how images change along multiple hyperparameter axes.

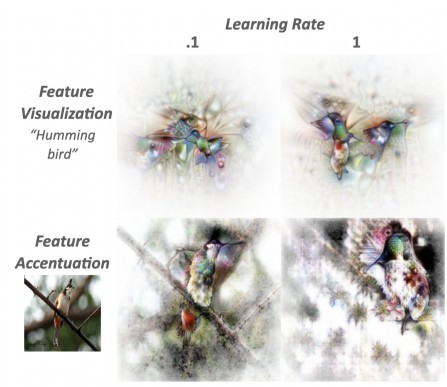

Figure 14: feature visualizations and accentuations for 'hummingbird' at two learning rates.

**Parametrization.** The Fourier and MACO parametrization, while undoubtedly more intricate, emerge as good contenders in generating meaningful perturbations. On the other hand, the Pixel parametrization method, while comparatively simpler in its approach, easily introduce adversarial pattern. The seemingly basic alterations to pixel values can yield strikingly impactful results on latent representation, akin to the subtleties found in adversarial perturbations.

**Augmentation.** With small crop augmentations, the regularizer and feature detector are essentially *zooming in* and making local edits to the image. Unsurprisely, this yields crisper accentuations. However, small crops can also lead to miniature hallucinatory features scattered throughout the accentuation, especially problematic given we seek explanations for the original, uncropped image.

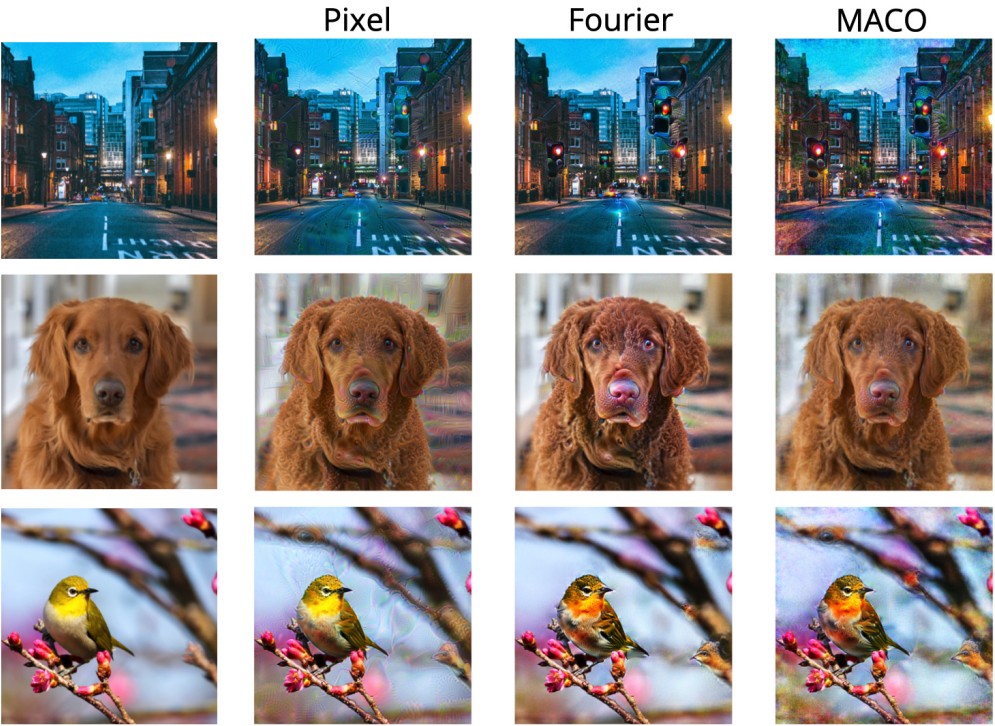

Figure 15: Additional example for our set of image parametrization.

Supplying a uniform mixture of crops each optimization step (as in the fourth column of Figure 16) seems to regularize against these hallucinations.

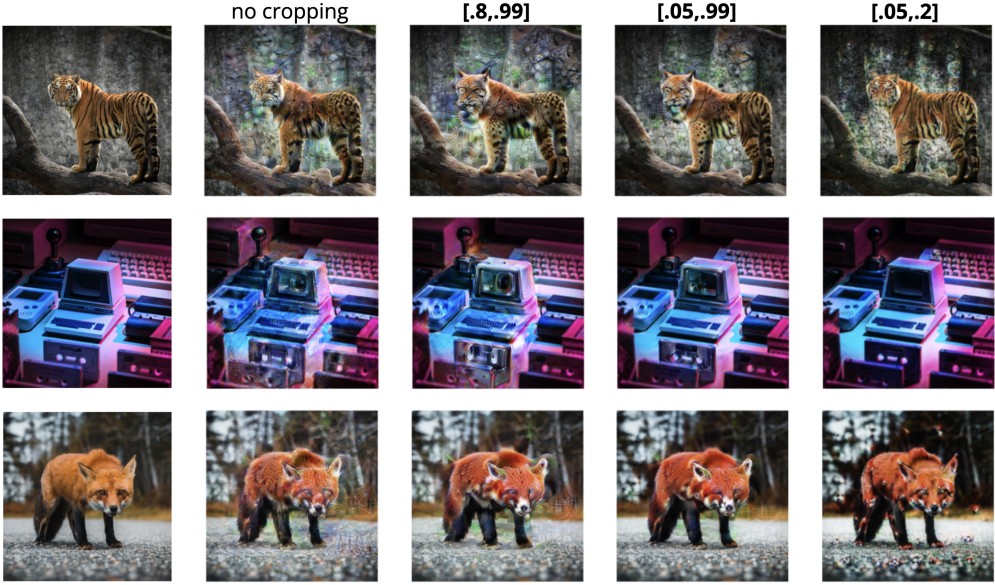

Figure 16: Additional cropping augmentation examples

**Regularization.** One of the major hyperparameter influencing our optimization process is $\lambda$ which serves as a regularizer. It plays a vital role in striking a balance between accentuation and preservation of the original data characteristics. More examples are shown confirming that a lower $\lambda$ value

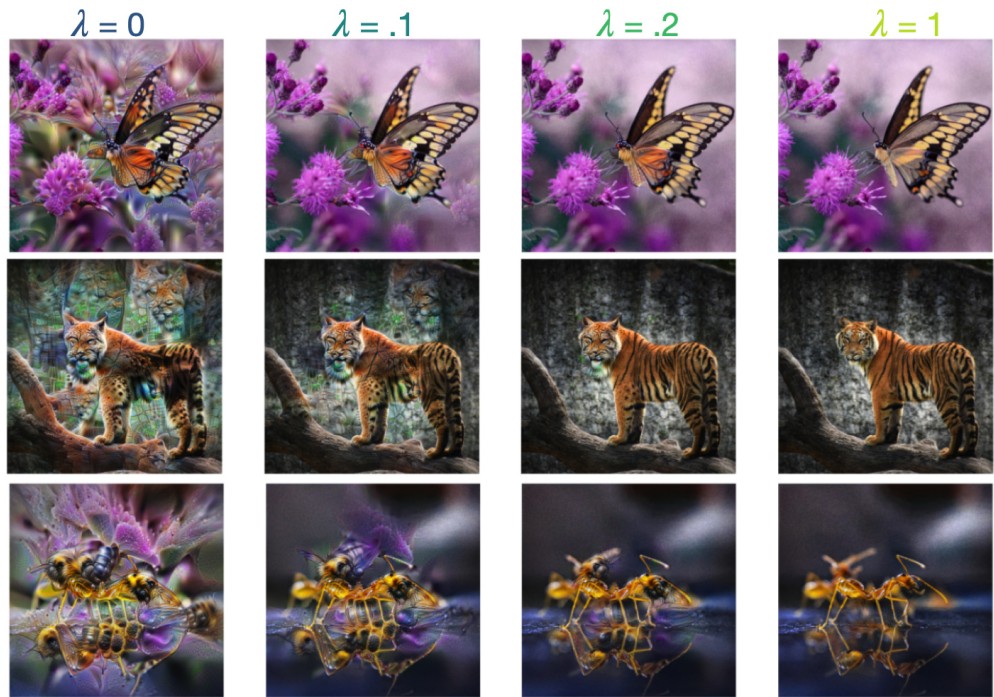

Figure 17: Additional regularization lambda examples

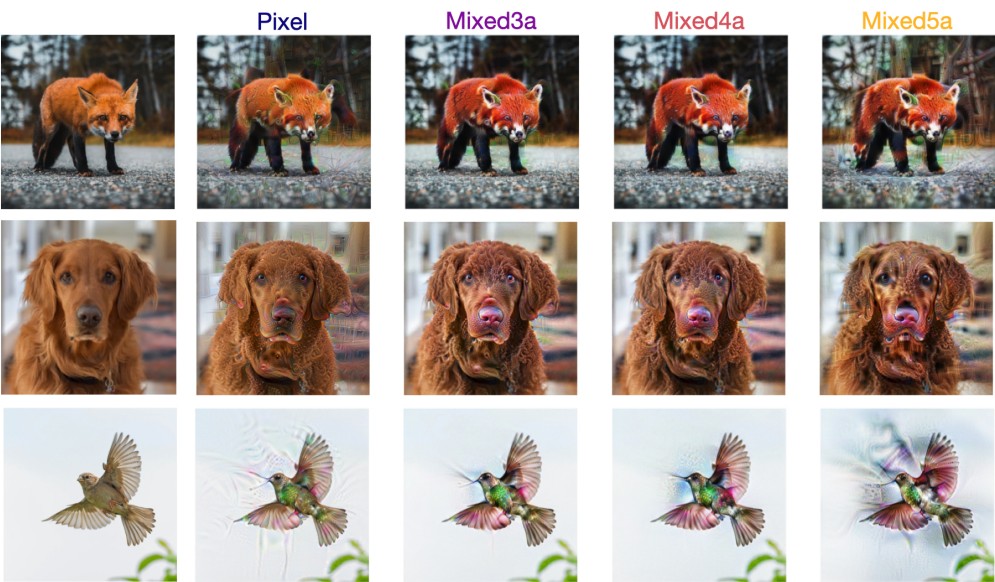

Figure 18: Additional regularization layer examples

can result in more pronounced accentuations, while a higher value tends to maintain the fidelity of the input data. We find the influence of $\lambda$ interacts with some other hyperparameters discussed here, namely the regularization layer and image parameterization. Given this, to facilitate a fair comparison between different parameterizations/regularization layers we generated accentuations at $\lambda = 0.05, .1, .5, 1, 5, 10$ for each, then chose the most natural looking image.

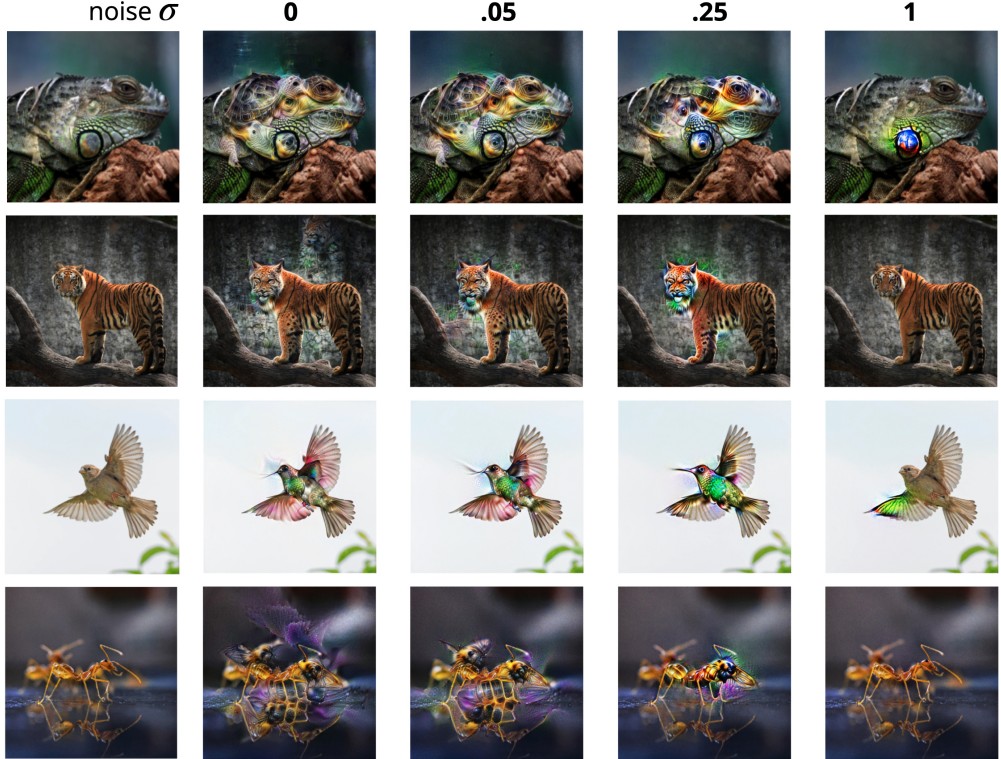

Figure 19: The effects of gaussian and uniform pixel noise as part of the augmentations $\tau$, given noise with different standard deviations $\sigma$.

**Effect of the Layer** Examining the impact of layer selection within our neural network architecture, we provide additional examples showing the effect of this hyperparameter on *Feature accentuation*.

The choice of the layer has a notable influence on the perturbation: different layers within our neural network exhibit distinct tendencies in accentuating specific features or patterns within the input data. Early layer tends to preserve pixel information while latter seems to allow greater perturbation but preserve semantic (and often class) information.

**Noise** To increase robustness of feature visualizations, Fel et al. (2023a) uses the addition of gaussian and uniform pixel noise as part of their augmentation scheme. The effects of such noise can be seen in Fig 22. We find that feature accentuations work well with no noise, thus we left this augmentation for the appendix. We find that when used in conjuction with or approach to regularization, adding noise tends to focus the areas in which feature accentuation augments the image. When a large amount of noise is applied the regularization term fully dominates and no perceptible changes are made to the image.

# E   RANDOMIZATION SANITY CHECK

Adebayo et al. (2018b) suggest a sanity check that any good explainability method should pass; an explanation for logit $y$ of image $x$ should perceptibly change when the network weights are randomized. This is because the function that returns $y$ from $x$, $f(x) = y$, is very different after weight randomization, thus it should require a different explanation. The authors show several attribution techniques fail this sanity check, here we check how feature accentuation fairs. Figure 20 shows feature accentuations under 'cascading randomization', where an increasing number of layers are randomization from top to bottom. We use the same 'junco' image used in the intial work, but also accentuate towards 4 different logits, those with the highest activation to the junco image (in the unrandomized model). We observe that while the original feature visualizations change the im-

age towards the targeted bird class, randomizing layers pushes the visualization towards seemingly random tagets. We use a $\lambda$ 0f .1 for every visualization, and observe that as additional layers are randomized the regularization term tends to dominate and the accentuations stop altering the image at all.

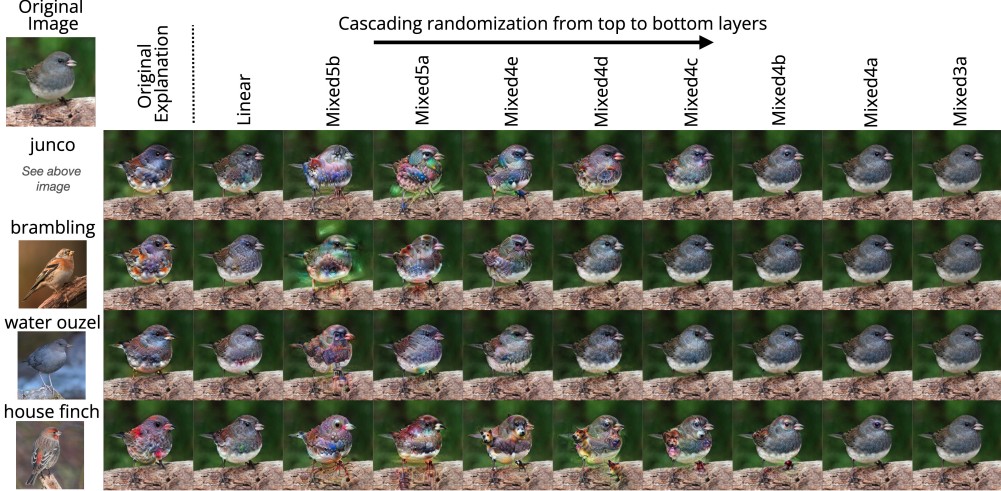

Figure 20: The 'cascading randomization' sanity check from (Adebayo et al., 2018b) performed on feature accentuation. From left to right: feature accentuation is performed on a model with more and more layers randomized. From top to bottom: different logit targets (corresponding to different bird classes) are used as the accentuation target.

## F  THE 'SENSITIVE REGION' FOR $\lambda$

While qualitatively evaluating the effects of regularization strength $\lambda$, we observe that for a given image $x$ and feature target $f_v$, proper accentuation requires $\lambda$ be within a certain 'sensitive region'. By this we mean, as lambda increases the resultant accentuation eventually becomes perceptually indistinguishable from the seed image, and conversely as $\lambda$ exponentially decays towards zero, the resultant visualizations becomes indistinguishable from the unregularized accentuation. Increasing $lambda$ within the intermediate region leads to visualizations that perceptually interpolate between the unregularized accentuation and the seed image. Here we consider how this sensitive region can be identified automatically, without visual inspection.

Such automation requires we establish a simple metric, measurable in the model, that corresponds with the perceptual changes induced by changing $\lambda$. The simplest candidate is the regularization term itself, $||f_\ell(x^*) - f_\ell(x_0)||$, that is, the distance between the representations of the accentuated image and the seed image in the regularization layer. In Figure 21.a we show for a single accentuation (iguana -¿ terrapin (turtle)) how changes in this distance correspond to perceptual change. As $\lambda$ becomes large/small this distance becomes fixed, as does the appearance of the accentuated image. This suggests the sensitive region can be found by first measuring the bounding distances for the accentuation from the seed image (at very small and very large $\lambda$), then searching for a $\lambda$ that yields a distance between these bounds.

It would be time consuming to search for $\lambda$ each time we want to run a new accentuation, but fortunately we find that for a given model layer the sensitive region is consistent across images/features. This can be seen in Figure 21.b; here we pass 50 random images through InceptionV1 and VGG11 (Szegedy et al., 2015), and accentuate each image towards its second highest logit across a range of regularization strengths. We plot $\lambda$ on the x-axis as before, and on the y-axis we plot *min-max normalized* the accentuation's distance from the seed image, where the minimums and maximums correspond to the maximum and minimum distances measured at extreme $\lambda$s $(10^{-6}, 10^6)$. We see that while each model requires a different setting for $\lambda$, there is a much smaller effect across images. As the sensitive region depends on the model and not the image/feature, for many use cases an ap-

propriate $\lambda$ must only be found once, and then many different examples can be generated using that setting, as we have done throughout this work.

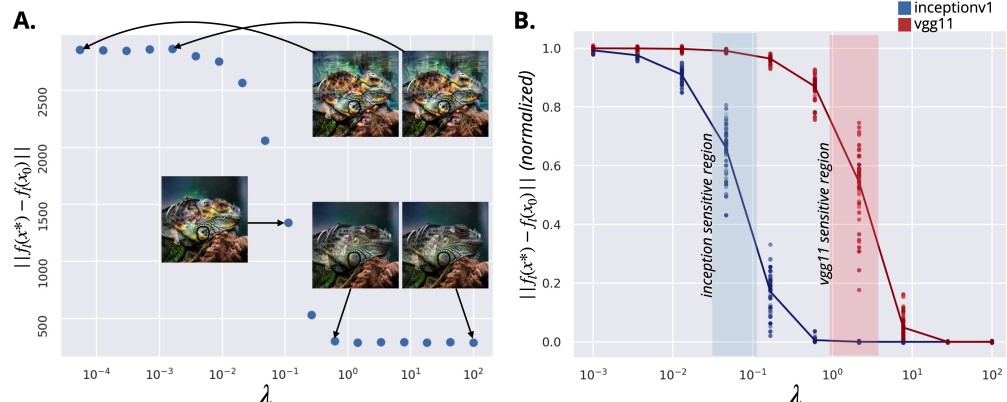

Figure 21: In **A.** we show the correspondence between the appearance of an accentuation at different $\lambda$ values and the distance of the accentuation from the seed image in the regularization layer. We note a *sensitive region* for $lambda$ in which changes perceptibly alter the level of accentuation. In **B.** we observe that the location of this sensitive region differs significantly across models, but not across images/features in a given model layer.

## G  FEATURE ACCENTUATIONS ON OTHER MODELS

Through this work we have focused on the InceptionV1 Mordvintsev et al. (2015) model given its prevalence in the feature visualization literature, but we find the general recipe put forth in this work is effective for other models as well. Fig **??** shows the effects of accentuations on several image/logit pairs for Alexnet (Krizhevsky et al., 2012), VGG11 (Szegedy et al., 2015), SqueezeNet (Iandola et al., 2016), and ResNet18 (He et al., 2016) models. Regularization $\lambda$ was set individually per model, to the 'sensitive region' as described in section F, and the regularization layer was set to the ReLU layer in each model closest to 1 model depth. All other accentuation hyperparameters are fixed across all examples to those hyperparameters described in section A.

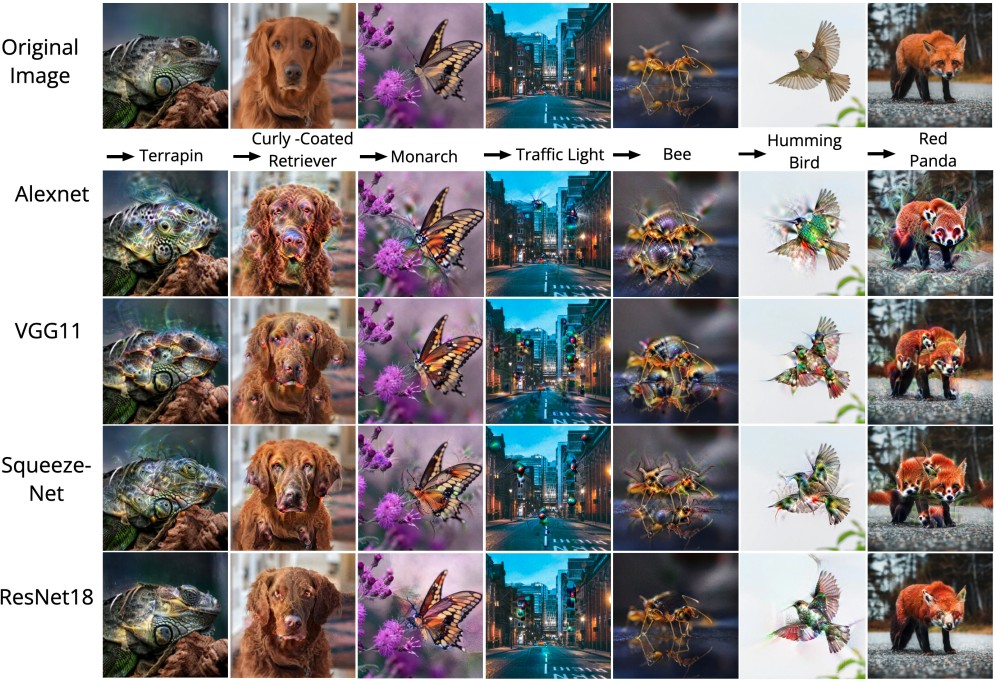

Figure 22: Feature accentuations towards logits for other popular Imagenet trained models.

