# OpenReview forum: "Feature Accentuation: Explaining 'what' features respond to in natural images"
_ICLR.cc/2024/Conference — Submitted to ICLR 2024_

### Official Review · Reviewer_bchW · 2023-10-14

**Soundness:** 3 good
**Presentation:** 4 excellent
**Contribution:** 2 fair
**Rating:** 6
**Confidence:** 3

**Summary:**

Previous methods focus on either where a model attends or what concept the model is looking for. This paper presents a method that can show both where and what a model is focused on -- this new method is called "feature accentuation". Feature accentuation is tested for natural-ness of images.

**Strengths:**

-  The paper is well-presented, along with a large number of visualizations littered both through the main text as well as the supplementary. The method is clearly explained in detail, as well as motivated explicitly each step of the way.
- The paper clearly proves what it sets out to -- namely, that the visualizations it produces highlight "where" and "what" in a natural-looking way.
- The evidence of natural-looking visualization is strong, and I'm convinced that feature accentuation can produce relatively realistic transformations, at least relative to previous deepdream-esque variants.

**Weaknesses:**

- My main concern is I'm not sure what the utility of feature accentuation is. A good chunk of the experiments focuses on natural-looking images, but I'm slightly less concerned about that. It's certainly a desirable property -- to look more natural and less like hallucinations. However, I'm not sure how this helps us, concretely: Can this help us improve accuracy? Diagnose mistakes? (This is suggested in several of the figures) Suggest a path for fixing the model? (Maybe by pruning "wrong" nodes?) There's a preview of this in figures 9 and 10, but the utility isn't studied in the paper. [1] for example has several approaches for defining and quantifying interpretability/utility of a method.
- Along the lines of the above, I'm not sure how to use the information these visualizations provide. For example, in figure 9, for "bow" vs. "chainsaw", I can certainly see that the "chainsaw" visualization is more chainsaw-like, but couldn't I just pull a random other word and visualize that? For example, I could feature accentuate "matchsticks" or "juggler fire torches", and I'm not sure how any of those visualizations would help me diagnose mistakes in the original model. Another way to put this would be: If I feature accentuate something clearly unrelated, eg.., "snail" for the "bow" image, it seems like this method would find *some way to insert a snail, but what does that tell me exactly, if I can insert any concept into the image? (One possibility is that natural looking insertions are "valid" explanations, and grotesque abstract art is unrelated? This would need some more fleshing out though, but that would be one way). On a side note, the main utility in figures 9 and 10 is ironically that it highlights some part of the original image, like saliency maps. (I'm not sure how important it is to be able to modify the image, per the above). A study would probably need to show that the image modification is helpful too.
- Have you tried the sanity checks in [2]? I know [2] is actually cited in your related works. Although feature accentuation is not a saliency map per se, the randomized tests in that paper should still apply. It would help (partially) my concerns above if the method does pass sanity checks -- in that it shows there *is meaning.

[1] Poursabzi-Sangdeh, et al. Manipulating and Measuring Model Interpretability. https://arxiv.org/abs/1802.07810
[2] Adebayo, et al. Sanity Checks for Saliency Maps. https://arxiv.org/abs/1810.03292

**Questions:**

I've left my questions above. In summary, I'm not convinced of the technique's utility, but I'm open to being convinced. The approach is certainly thoroughly explored and visually appealing. I'm just afraid that visual appeal could be a misleading objective.

---

> ### Author Response · Authors · 2023-11-17
>
> We thank the reviewer for taking the time to read our article and for the valuable feedback.
>
> **“My main concern is I'm not sure what the utility of feature accentuation is.”**
>
> Feature Accentuation represents a novel tool that, as you rightly pointed out, yields feature visualizations ranging from more to less naturalistic. In recent years, general feature visualizations have significantly impacted the literature, enabling the understanding of internal mechanisms within neural networks (such as specific filters and invariances)[1,2,3], detection of biases [4], enhanced comprehension of complex concepts [6], and visualization of internal states of intricate models[7].
>
> We have specifically chosen to illustrate the utility of our method with a use-case presented in Figure 9. This example demonstrates how Feature Accentuation can provide valuable insights in explaining failure cases. For instance, in the first image, it might seem perplexing that an image of a barber shop is classified as a moving truck. Our approach sheds light on such cases revealing that the model mistakenly associates the white sheet on the wall with the trailer of a truck.
>
> We posit that Feature Accentuation serves as a superset of traditional feature visualizations, offering the potential to accelerate all those use cases simultaneously. By providing a way to explore a diverse set of explanations ranging from a natural image to purely abstract feature visualization, our method contributes to the broader landscape of interpretability tools.
>
>
> **“couldn't I just pull a random other word and visualize that? For example, I could feature accentuate "matchsticks" or "juggler fire torches", and I'm not sure how any of those visualizations would help me diagnose mistakes in the original model.”**
>
> This is an excellent point, and something that concerned us as we were developing the method. Indeed, if we can accentuate arbitrary features, it’s possible the technique could mislead the user, ‘revealing’ how features are expressed in an image when in fact there is nothing in the image that excites the feature. We demonstrate this phenomenon empirically in Figure 6. Given this concern, we demonstrate in section 2.4 how feature accentuations can be combined with attribution masks as a means of filtering, revealing exaggerated versions of the original image only in those image regions that excited the feature in the original. In this way, feature accentuation can help intuit the many different ways natural images excite a feature, without suggesting to the user that everything excites it.
>
> **“Have you tried the sanity checks?”**
>
> Thank you for this suggestion. We have implemented the randomization sanity check from the paper, and our method successfully passes it. We’ve added the results to the appendix (section E).
>
> [1] Cammarata, et al., "Curve Detectors", Distill, 2020.
> [2] Olah, et al., "Naturally Occurring Equivariance in Neural Networks", Distill, 2020.
> [3] Schubert, et al., "High-Low Frequency Detectors", Distill, 2021.
> [4] Singla, et al., “Core Risk Minimization using Salient ImageNet”, ICLR 2022
> [5] Fel, et al., “CRAFT: Concept Recursive Activation FacTorization for Explainability”, CVPR 2023
> [6] Ghiazi, et al., “What do Vision Transformers Learn? A Visual Exploration”

---

> ### Comment · Reviewer_bchW · 2023-12-03
> **Thanks for the clarifications**
>
> - The sanity check increases my confidence in the paper; thanks for running those checks.
> - Re 2.4: This seems reasonable. Combining feature accentuation with attribution masks is a possibility, and I can certainly see it mitigating accentuation of random features. (But does the need for one mean that feature accentuation is somehow deeply flawed? You could conversely argue that these two are orthogonal visualization methods that are stronger--and only sensible--when combined)
> - I'm still concerned about the utility of the method. Whereas I could intuit the utility of feature accentuation, there are more concrete ways of defining and proving interpretability; [1] above was recommended to me before, which accomplishes this -- mainly by quantifying utility through improved human/model performance on certain tasks. It's currently hard to assess the evidence for feature accentuation's utility, besides just a visual "yep looks good"; for example, I can't claim feature accentuation has been shown to improve debug-ability in some quantifiable way.
>
> After reading through the other reviews and their concerns, it seems like most concerns have been addressed; I've bumped my score up.

---

### Official Review · Reviewer_L6NW · 2023-10-30

**Soundness:** 4 excellent
**Presentation:** 2 fair
**Contribution:** 4 excellent
**Rating:** 8
**Confidence:** 3

**Summary:**

This paper proposes a new method for generating visualisations of the features leading to specific activity of neural network models, termed Feature Accentuation.
The primary novelty here is to optimise a new image such that a weighted tradeoff is generated between maximising the activation of the unit under study while staying close to the original image.
Various techniques are used (image parameterization, augmentation) to make this work.
The results of this process look generally appealing, and for certain examples quite compelling, in showing how the technique can swap class labels with subtle or not-so-subtle but still sensible image changes.
It is also appealing that the technique can be applied without needing to use an auxiliary generative model.

**Strengths:**

My initial recommendation is to accept the paper, since it appears to solve (or at least, offer paths to solution) for a core problem in explainability / network visualisation.
I have some minor comments on the paper.
However, I must caveat this by saying that I have not worked in this area for several years, and so my knowledge of the literature is outdated.
It is possible that other reviewers are aware of work that undermines the novelty or contribution of this approach.

**Weaknesses:**

- The paper is sloppily formatted (parentheses missing from citations, missing references, etc).
- Unpack the equation for $z^{*}$ (top of page 4) into words, since it's the key equation of the paper.
- raster plots in figure 8 are poor quality; hard to see detail.
- Figure 8 B, C: lambda should probably be 1.0 not 10.0
- Figure 10 caption should make explicit what the rows and columns are.

**Questions:**

- How novel is the frequency domain parameterisation? citations to other literature should be provided.
- What does it mean to achieve higher correlations in circuit similarity than natural images themselves? In a positive view, this could mean that the feature accentuation technique is settling onto good *prototypes* that provide a coherent illustration of the core concept of the label, and thereby reduces variance from natural depictions of the concept. A less positive view could be that this means the technique is finding local minima that will fail to generalize.
- Figure 8: what are the smooth curves, and how were their hyperparameters chosen? The underlying data are very noisy, so the choice of smoothing and its associated uncertainty should be reported.

---

> ### Author Response · Authors · 2023-11-17
>
> We are delighted to hear that you found the paper enjoyable, and we extend our sincere gratitude for investing time and effort in providing a thorough review. We address some of your concerns below.
>
> **Formatting, references, image/figure quality**
>
> Thank you for your feedback; we have incorporated it into the manuscript. As our paper contains a lot of images we were constrained by the file size limit on our submission, and subsequently compressed the PDF too aggressively. Our undated manuscript is higher resolution, and the camera-ready version will feature the highest possible quality.
> How novel is the frequency domain parameterisation? citations to other literature should be provided.
> While the frequency domain parameterization is not a novel concept and has been discussed in [1,2], we have, nonetheless, integrated and tested a more recent parametrization (NeurIPS 2023) that proves effective on newer models [3]. Additionally, we have included several missing citations.
>
> **"What does it mean to achieve higher correlations in circuit similarity than natural images themselves? In a positive view, this could mean that the feature accentuation technique is settling onto good prototypes that provide a coherent illustration of the core concept of the label, and thereby reduces variance from natural depictions of the concept. A less positive view could be that this means the technique is finding local minima that will fail to generalize."**
>
> Our interpretation aligns with your ‘positive view’, which you articulate quite well (perhaps better than we articulate ourselves in the paper)! With regards to generalization, we are confident this result will generalize to new examples given how it was calculated. Each natural image in our experiment can be paired one-to-one with an accentuation of that image towards its class logit. These accentuations are conditioned in no way on any other images in the experiment. Within a given class, we get the average pair-wise correlation for each natural image, and similarly the average correlation of each accentuation to the natural images (excluding the correlations between accentuations and their natural image pair, which would be trivially high). We find the accentuations correlate higher on average, which means accentuating an image towards its class tends to make it follow a circuit closer to other, randomly selected images of the class.
>
>
> **''Figure 8: what are the smooth curves, and how were their hyperparameters chosen? The underlying data are very noisy, so the choice of smoothing and its associated uncertainty should be reported.''**
>
> The curves depicted in these figures are a spline interpolation (degree 2), across the underlying data-points (class-wise average correlations) averaged into 10 bins. We took this approach so as to convey the raw data and the general trend simultaneously. A difficulty with plotting this data stems from the fact that there is a large amount of variance in the correlation measure across nearby layers of different architectural type. We can filter out this variance in Figure 8.a and isolate the effect we are interested in by plotting the difference between the correlation measure for accentuations and natural images in each layer. We have added this version of the plot – which requires no smoothing – to appendix A, as well as additional clarification on the figures and design of the path coherence experiment.
>
> **“Figure 8 B, C: lambda should probably be 1.0 not 10.0”**
>
> We actually meant for this to be 10.0, as we wanted to measure the effects setting lambda to extreme values.
>
> [1] Differentiable Image Parameterizations, Mordvintsev, et al. Distill, 2018.
> [2] Feature Visualization, Olah, et al. Distill, 2017.
> [3] Unlocking Feature Visualization for Deeper Networks with Magnitude Constrained Optimization, Fel, et al. Neurips 2023.

---

> > ### Comment · Reviewer_L6NW · 2023-11-22
> > **Review score unchanged**
> >
> > I thank the authors for their replies to my questions. I have read the replies and the other comments. My review score will remain unaltered.

---

### Official Review · Reviewer_kHZK · 2023-11-01

**Soundness:** 2 fair
**Presentation:** 2 fair
**Contribution:** 2 fair
**Rating:** 6
**Confidence:** 3

**Summary:**

The paper introduces a model explanation technique that aims to amplify image features for a discriminative model. Specifically, the method finds visualizations that resemble an input image and a target feature, without relying on external models. The paper demonstrates the effectiveness of the framework via qualitative examples and showcases multiple applications.

**Strengths:**

* The approach is well-motivated and well-explained, starting from an overall objective and then the necessary additional components to address challenges encountered.
* The fact that the proposed approach does not rely on external models makes the framework a standalone explainability framework that probes the internal knowledge within one model only.

**Weaknesses:**

* All results in the main paper are quantitative and are shown only on a few selected examples.
* The framework is sensitive to hyperparameters such as learning rates and regularization weights, as noted in Appendix C-D, which makes it challenging to adapt to different models.

**Questions:**

* Multiple qualitative examples are shown but the analysis is lacking. For example, how to interpret the results from Figure 12 in the Appendix?

---

> ### Author Response · Authors · 2023-11-17
>
> Thank you for your review, you raise some important concerns that we address below.
>
> **“All results in the main paper are quantitative (assume means qualitative)”**
>
> It’s true that a significant portion of our results focuses on qualitative outcomes, as in some sense feature accentuation is an intrinsically qualitative explanation method; it is designed to produce visualizations a user can qualitatively evaluate to gain intuition for what excites a given feature about an image. Attribution methods and feature visualizations, the standard approaches in this area, are also qualitative in this respect. That said, we respectfully disagree with the notion that our paper lacks quantitative depth. We would like to draw attention to an entire section dedicated to quantitative results (Section 3, circuit analysis, Figure 8 a, b, and c) that we feel compelled to highlight.
> In this specific section, we argue, akin to the findings presented in [1], that feature accentuations for maximizing an activation should do so in a ‘natural’ way, i.e. by a similar path through the network as the natural images that excite the neuron. Quantitatively, we demonstrate that our method – which does not require additional models that enforce a natural-image prior – achieves excellent results when compared to state-of-the-art feature visualizations on this metric (Figure 8.a). Furthermore, we argue this analysis can be extended to address whether feature accentuations are providing veridical, local explanations for their seed image. That is, a feature accentuation is not a good explanation for why a neuron is excited by an image if the accentuation takes a distinct path through the network. Figure 8.c shows that increasing regularization strength keeps the path through the entire network closer to the seed image. The quantitative evidence presented in Section 3, coupled with our method's independence from additional models, make this work an advancement on the original feature visualizations. We believe this unique aspect of our methodology not only adds depth to the interpretability toolkit but also addresses concerns raised by previous studies.
>
> **“The framework is sensitive to hyperparameters such as learning rates and regularization weights, as noted in Appendix C-D, which makes it challenging to adapt to different models.”**
>
> Indeed, our framework includes a regularization term designed to keep accentuations close to the seed image while performing activation maximization for a target feature. We posit that this hyperparameter doesn't necessarily require tuning, but rather allows the user to control the extent to which the target feature is exaggerated, a desirable property for VCEs [2]. Additionally, we have added a small experiment to the appendix (section F) that further explores the setting of regularization strength in different contexts. We find that while the appropriate level of regularization must be identified across models, the same regularization strength works well across images/features within a model layer. Further, we present a method for finding the appropriate amount of regularization that does not require visual inspection by the user, thereby automating the selection of this hyperparameter.
> Regarding other hyperparameters, while we do spend a considerable portion of the paper exploring the hyperparameter space, we ultimately prescribe settings that work well in the general case. With regards to the learning rate in particular, our phrasing in appendix C was too strong (it has been updated), a more accurate statement is that the upper limit for the learning rate in feature accentuation is lower than for feature visualization. As long as the learning rate is sufficiently low, it will work well in a variety of contexts. To demonstrate this, we have added Appendix F, which shows feature accentuations for a range of models. The only hyperparameter that is adjusted in this experiment is the regularization strength, which is set once per model, but not adjusted across different images. All other hyperparameters are fixed across all examples. The regularization layer, which of course can not be identical across models, is simply set to the ReLU layer closest to ¼ depth through each model.
> “how to interpret the results from Figure 12 in the Appendix?”
> An interpretation of ‘super-natural images’ like those depicted in Figure 12, can be found in Section 3; they are images generated by feature accentuation that take a prototypical path through the network for their corresponding class. We put Figure 12 in the appendix to limit the number of images in the main paper (as there are a lot already). Still, we presumed the reader would be curious to see what these ‘super-natural’ images actually looked like. We have clarified the figure caption.
>
> [1] Don't trust your eyes: on the (un)reliability of feature visualizations, Geirhos et al., 2023
> [2] Sparse visual counterfactual explanations in image space, Boreiko et. al.  2022

---

> > ### Comment · Reviewer_kHZK · 2023-12-05
> > **Post-Rebuttal Responses**
> >
> > Thank you authors for the responses on quantitative evaluation and hyperparameter tradeoffs. I've read the rebuttal responses and comments from other reviewers.
> > 1. Regarding quantitative evaluation, I should have clarified that I think there lack a quantitative measure towards the final goal of the proposed method, e.g., similarity of the feature visualizations with the target class, or the usefulness of the method in downstream tasks such as explaining misclassified images. Figure 8A compares the method with natural images, MaCo, and Fourier, which serves more similar to ablation studies as these are different parameterization choices if I'm understanding it correctly. More explanations on the baseline methods would help clarify. And this metric only measures the circuit similarity aspect.
> > 2. Regarding the choice of $\lambda$, the authors have addressed my concern.
> >
> > There are some other details that can be clarified in the revised version of the paper:
> > 1. In Figure 3., in the bottom row, it's not explained which $\lambda$ value is chosen.
> > 2. In Figure 9., it's not straightforward to see that "barber shop" refers to the incorrect prediction output, and the image below is the input, and the "moving truck" is the target feature, and the image below is the accentuated feature map.
> >
> > I am convinced about the novelty and potential of usefulness of the proposed work. I've raised my score accordingly. My remaining concerns are that some details as mentioned above should be further clarified, and some claims, e.g. the usefulness of cropping augmentation mentioned in Section 3.2, can be analyzed with a few more examples in the revised version of the paper.

---

### Official Review · Reviewer_zCew · 2023-11-03

[review text omitted: it was posted to a different submission]

---

> ### Author Response · Authors · 2023-11-17
>
> Unfortunately, the written review pertains to another article; the abstract is different from our paper.

---

> > ### Comment · Reviewer_zCew · 2023-11-20
> >
> > I would like to sincerely apologize for the mismatch review.
> > I write here the one that was originally written for this paper:
> >
> > Summary:
> > This paper introduces a method called feature accentuation. It is a new explainability method that it indicates which pixels in the image are relevant for the final decision of the model. As well as which kind of features activate relevant neurons.
> >
> > Soundness: 3 good
> > Presentation: 3 good
> > Contribution: 2 fair
> > Strengths:
> > A strength of this method is not needed auxiliary generative models and being seeded to images. Also, the release as an open-source library as part of Lucent will make it accessible to those who would like to use it for their applications or built on top of it.
> >
> > To overcome some gaps from previous related research, this methods incorporates several techniques to avoid that the modified image that accentuates some feature changes how it activates the neuron’s with respect to the original seed image as a regularisation term in the loss. There is an analysis on which layers play an important role to avoid undesired distortions, and it is reported that enforcing it in earlier layers yields better visualizations in early layers. An analysis in the impact of regularisation, parametrisation and augmentation techniques from the literature applied in this method is conducted, highlighting the right combination of those factors. Additionally, to improve relevance of the feature representations, a global normalisation is proposed
> >
> > The experiments reported use circuit coherence assessment from another paper, which is a reasonable measure.
> >
> > Weaknesses:
> > The other applications showcased are also of high importance, but the results become more difficult to assessed. How confirmation biased is overcomed with this method? It is still based on visualisations which need human assessment. The what is based on visual information that is difficult to parse for a human. How useful then it really is still an open question. This is already mentioned in limitations, but it is a strong self-critic that should be given more thought on how to overcome those. How to in corporate over tools for the interpretation is also not clear.
> >
> > MINOR: There is a reference missing with a question mark. There are a couple of blank space missing to segment words.
> >
> > Questions:
> > Please refer to weakness points.
> >
> > Flag For Ethics Review: Yes, Other reasons (please specify below)
> > Details Of Ethics Concerns:
> > It's a minor concern. It is not clear how useful is the explainability with this tool.
> >
> > Rating: 6: marginally above the acceptance threshold
> > Confidence: 4: You are confident in your assessment, but not absolutely certain. It is unlikely, but not impossible, that you did not understand some parts of the submission or that you are unfamiliar with some pieces of related work.
> > Code Of Conduct: Yes

---

> > > ### Author Response · Authors · 2023-11-21
> > >
> > > Thank you for taking the time to provide an updated review! We appreciate your thorough evaluation. However, could you please include your updated score for the paper in the official scoring section? Currently, it's only mentioned in the previous comment.
> > >
> > > **'How is confirmation bias overcome?'**
> > >
> > > We agree that confirmation bias is an important hurdle explainability methods should strive to overcome, and feature accentuation is not wholly immune to it. However, when compared to similar approaches our method offers some advantages that make confirmation bias less likely. For example, a VCE generated with the guidance of an auxiliary generative model [1] will look very natural, which may instill confidence that the explanation is veridical. However, the generative model is enforcing a prior that the VCE must look natural, maybe the model under investigation saw something quite different in the original image. Conversely our method synthesizes an optimal image through self-guidance from the discriminative model, there is no constraint that the result should look natural to a human. Furthermore, our results in Figure 8.c demonstrate our regularization technique pushes accentuations to be processed in a similar way to the seed image by the model. This constitutes additional evidence that feature accentuations are actually telling us something about the model, rather than just showing images that are perceptually intuitive.
> > >
> > > **'[Feature accentuation is] Based on visual information that is difficult to parse for a human'**
> > >
> > >
> > > Our technique is one of many that uses visual information to explain a vision model. We contend that one of the strengths of feature accentuations using our implementation is they are often quite intuitive for a human to parse. Consider, for example, the results we obtain in Figure 9, where we accentuated those features which lead to misclassifications. In each example accentuation makes the misclassified feature clear in the image, in such a way that the hallucination can be better perceived in the unaltered original image as well. We contend that using attribution maps or conventional feature visualizations could not resolve such examples so cleanly.
> > >
> > >
> > > **'How to incorporate other interpretability tools is not clear'**
> > >
> > > We comment on the importance of other explainability tools so as to not suggest accentuation is a universal solution for explainability. It is not necessarily possible to incorporate it directly with every other method, rather it is additional to the XAI toolbox, which can provide unique insights. That said, there are other tools with which feature accentuation naturally integrates. We demonstrate explicitly in the paper how attribution methods – a popular family of techniques – can be integrated with feature accentuation. Additionally, concept-based approaches [2][3][4], which seek to identify human interpretable directions in a model’s latent space, naturally integrate with feature accentuation; concepts can be treated in the same manner as ‘channels’ (figure 10), my maximizing the dot product with the concept direction. This functionality is implemented in our faccent API.
> > >
> > >
> > > **'Citation missing, minor edits, etc.'**
> > >
> > > Thanks for catching this, we will update the manuscript accordingly.
> > >
> > >
> > >
> > > [1] Augustin et. al. “Diffusion visual counterfactual explanations” (2022) Neurips
> > >
> > > [2] Kim et. al. “Interpretability beyond feature attribution: Quantitative testing with concept activation vectors (TCAV)”, ICLR (2018)
> > >
> > > [3]Gesina Schwalbe, “Concept Embedding Analysis: A Review” (2022) arXiv:2203.13909
> > >
> > > [4] Fel et. al. “Craft: Concept recursive activation factorization for explainability, CVPR (2023)

---

### Author Response · Authors · 2023-11-17

We’d like to thank all the reviewers for their time and thoughtful comments, we have responded to each in individual responses below. Additionally, we have posted an updated version of the manuscript, with added appendices E, a sanity check experiment (as suggested by reviewer **bchW**),    Appendices F and G which explore the setting of hyperparameters across different models (addressing concerns raised by reviewer **kHZK**), clarification of details in our path coherence experiment and the associated plots (reviewer **L6NW**) as well as general formatting/cosmetic updates.

---

### Meta-Review · Area_Chair_31vV · 2023-12-11

**Metareview:**

In this paper the authors introduce a method for model explainability in image classification models. In particular, the authors employ an optimization method to iteratively refine an image to accentuate either the top predicted class or any other class that the model might predict. The authors demonstrate the quality of these visualizations using convolutional neural networks such as VGG and Inception-v1. The visualizations provide visually striking qualitative examples about how an image may be interpreted as, say, other potential classes. One positive aspect of the method is that the authors show examples of how failed predictions may be interpreted based on the visualizations. The reviewers commented positively on the simplicity of the method, and the lack of requiring an external model. The reviewers also highlighted significant concerns about the overall approach, in particular (1) the fact that the visualizations are difficult to interpret and may lead to confirmation bias, (2) the difficulty of incorporating the work into other interpretability approaches, and (3) the lack of quantitative evaluations. The reviewers were satisfied with the author's responses to each of these points, however the AC found these explanations unsatisfactory.

The issue of confirmation bias is critical and speaks to a central scientific challenge missing in this paper. Given how difficult it is to judge whether a given visualization appropriately “explains” a classification, it is very easy to arrive at answers that merely reinforce an (incorrect) intuition for the performance of a model. One natural way to resolve this issue is to perform a battery of human evaluation experiments (psychophysics) to quantify statistically how and when these visualizations are consistently judged by human perception, and argue how this correspondence is superior to other methods based on such a quantification. I should note that equally interesting in such experiments is when the proposed method does not match human perceptual judgements as such a result additionally showcases opportunities for methodological improvements.

Unfortunately, attempting to quantify how and when such visualizations correspond to human perception is entirely lacking in this work. Without such a quantification, the visualizations – while aesthetically appealing – do not represent falsifiable science. Although this work will not be accepted to this conference, I suspect that if the authors endeavor to invest a serious effort in measuring human perceptual judgements, this would add strong statistical evidence about the utility of the proposed interpretability method.

**Justification For Why Not Higher Score:**

Qualitative results. Not falsifiable and subject to confirmation bias.
One reviewer mistakenly scored 3 instead of intended 6.

**Justification For Why Not Lower Score:**

N/A

---

### Decision · Program_Chairs · 2024-01-16

Reject